# Analysis of the Association between Retinal Artery Occlusion and Acute Ischaemic Stroke/ST-Elevation Myocardial Infarction and Risk Factors in Hungarian Patients

**DOI:** 10.3390/medicina59091680

**Published:** 2023-09-18

**Authors:** Szabolcs Balla, Attila Vajas, Orsolya Pásztor, Anikó Rentka, Balázs Lukucz, Márta Kasza, Attila Nagy, Mariann Fodor, Valéria Nagy

**Affiliations:** 1Department of Ophthalmology, University of Debrecen, 4032 Debrecen, Hungaryorsolyapasztor3@gmail.com (O.P.); rentka.aniko@gmail.com (A.R.); mfodor@med.unideb.hu (M.F.); valnagy1@gmail.com (V.N.); 2Department of Technology and Economics, University of Budapest, 1111 Budapest, Hungary; lukuczb@gmail.com; 3Medical Centre, Hungarian Defence Forces, 1134 Budapest, Hungary; kaszam77@gmail.com; 4Department of Public Health and Epidemiology, Faculty of Medicine, University of Debrecen, 4032 Debrecen, Hungary; attilanagy@med.unideb.hu

**Keywords:** retinal artery occlusion, stroke, secondary prevention, role of ophthalmologists

## Abstract

*Background and Objectives*: We aimed to analyse data on retinal artery occlusion (RAO) patients to explore correlations with acute ischaemic stroke (AIS), ST-elevation myocardial infarction (STEMI), and cardio/cerebrovascular comorbidities. *Patients and Methods*: Our retrospective cohort study included 169 RAO and 169 age- and gender-matched control patients. We examined the association of AIS, STEMI, and related comorbidities such as hypertension (HT), type 1 and type 2 diabetes (T1DM and T2DM, respectively), hyperlipidaemia, and ischaemic heart disease (IHD) with RAO. We also recorded atrial fibrillation in our RAO patients. *Results*: Our results demonstrated that RAO patients developed both AIS and STEMI at a significantly higher rate compared to controls (*p* < 0.001 for both). We also found that RAO patients had a significantly higher prevalence of HT and hyperlipidaemia (*p*1 = 0.005, *p*2 < 0.001) compared to controls. Multiple risk factors together significantly increased the odds of developing AIS and STEMI. *Conclusions*: Our results suggest that through identifying and treating the risk factors for RAO patients, we can reduce the risk of AIS, STEMI, and RAO of the fellow eye. Considering that ophthalmologists are often the first detectors of these cardiovascularly burdened patients, collaboration with colleagues from internal medicine, cardiology, and neurology is essential to achieve secondary prevention.

## 1. Introduction

The global population and average life expectancy at birth continue to increase over time. In addition, acute cardio- and cerebrovascular mortality is decreasing as the quality of care for ST-elevation myocardial infarction (STEMI) and cerebrovascular accidents (stroke) improves. However, cardio- and cerebrovascular events remain the leading causes of death globally [1,2,3,4].

Retinal artery occlusion (RAO) is a vascular disorder that affects the arterial network that supplies the inner two-thirds of the neuroretina [5,6]. This painless event usually results in immediate and often irreversible visual impairment [7,8,9]. Figure 1 illustrates the most common RAO types, central retinal artery occlusion (CRAO) and branch retinal artery occlusion (BRAO) [10].

Cilioretinal arteries are an anatomical variant that occurs in 10–15% of people. These branches originate either directly from the choroid or from one of the posterior ciliary arteries. In those with such an arterial branch, in cases of CRAO, it may provide the minimum blood supply required to the macula lutea, resulting in less marked visual impairment. Inversely, these patients may have a rare form of arterial branch occlusion, i.e., cilioretinal artery occlusion. These branches arise either directly from the choroid or from one of the posterior ciliary arteries [11]. There are two forms of RAO, the more common being non-arteritic and the less common arteritic form (giant cell arteritis). If RAO develops, the arterial form should be ruled out as a matter of urgency, as it can compromise vision in the fellow eye or affect the blood vessels of other vital organs. The arteritic form can be excluded via testing for elevated inflammatory biomarkers such as erythrocyte sedimentation rate (≥47 mm/h) and C-reactive protein (CRP) (≥2.45 mg/dL). In addition to laboratory examination, the presence of clinical signs such as fever, chewing muscle pain, headache, scalp-like pains, and muscle pain may help in the diagnosis [12]. In cases of strong clinical suspicion, temporal artery biopsy may also be considered. The arteritis form requires emergency management with high-dose intravenous steroids. Treatment with megadose steroids may prevent severe bilateral visual loss and systemic complications. The non-arteritic form is essentially a retinal infarct. Initially, the retinal ganglion cell layer is affected by hypoxia, followed by progressive axonal degeneration, not only in the retinal cells but also in the optic nerve. Acute ischaemic stroke (AIS) is an episode of neurological dysfunction caused by a blood clot that blocks an artery leading to the brain [13,14]. STEMI occurs due to the occlusion of coronary arteries (one or more), causing transmural myocardial ischaemia with persistent electrocardiographic ST elevation, which can result in myocardial damage or necrosis [15,16]. The risk factors for RAO, AIS, and STEMI are almost identical, including advanced age, male gender, hypertension (HT), type 1 or 2 diabetes mellitus (T1DM/T2DM), hypercholesterolemia, atrial fibrillation (AF), ischaemic heart disease (IHD), tobacco use, and obesity [17,18,19,20].

Our retrospective cohort study aimed to investigate the association between RAO and the development of AIS and STEMI. We investigated the role of age and gender in the development of RAO, AIS, and STEMI in Hungarian patients. We also investigated the role of cardio/cerebrovascular risk factors in the development of RAO, AIS, and STEMI in our study patients. We aimed to analyse the combined effect of several risk factors on RAO, AIS, and STEMI development in Hungarian patients. We sought to determine whether treatment or reduction of risk factors could reduce the incidence of RAO, AIS, and STEMI (secondary prevention).

## 2. Patients and Methods

### 2.1. Patients

Data on 181 RAO patients (114 CRAO and 67 BRAO) were examined at our clinic in the Eastern European region between 2009 and 2019. Data were retrospectively collected from our electronic database and the Hungarian national electronic database.

A total of 169 RAO patients (106 CRAO and 63 BRAO) were retained for analysis. Risk factors were considered if they were recorded in the documentation. Comorbidities were recorded based on clinical data and the regular medications provided.

### 2.2. Methods

Our study included adult patients with visual loss due to RAO within 1 week. The inclusion criteria comprised an RAO diagnosis based on anamnestic data (sudden vision loss or visual field defect) and characteristic fundus findings of diagnostic value (pale, oedematous retinal layer with a cherry-red spot in the foveolar area, sludging or embolus in the arterioles).

Our exclusion criteria were the presence of other ocular disorders (e.g., age-related macular degeneration), RAO associated with other retinal vascular disorders (e.g., thrombosis, non-arteritic anterior ischemic optic neuropathy), suspected iatrogenic RAO (e.g., fat embolisation following open heart surgery). RAO patients whose electronic documentation was not accessible due to digital self-determination or who did not have adequate documentation of concomitant illnesses or medications used were also excluded from the study. For these reasons, 12 patients had to be excluded from the study.

The patients’ age, gender, and RAO onset and type (CRAO/BRAO) were documented. The causes of death were analysed with particular regard to cardio- and cerebrovascular diseases. We also recorded whether patients had AF. Additionally, the temporal relationship between cardio/cerebrovascular death and RAO development was investigated.

A total of 169 age- and gender-matched controls were selected from among hospital patients. The patients who were about to undergo cataract surgery and who did not have any retinal vascular disease in the electronic database were selected as controls. All preoperative controls underwent a complete ophthalmic examination in our clinic before surgery (best corrected visual acuity, slit lamp examination, intraocular pressure measurement via Goldmann applanation tonometry, funduscopy with dilated pupils). For both groups, the presence of AIS, STEMI, and concomitant diseases, such as HT, T1DM, T2DM, atrial fibrillation (AF), hyperlipidaemia and IHD, as well as the time of onset, were recorded.

### 2.3. Statistical Analysis

Descriptive statistical analysis was performed comparing patient numbers, age, gender, AIS, STEMI, and comorbidity distribution between the RAO and control groups. Normality was evaluated using the Shapiro–Wilk test. Controls matched for age and gender were used, and the homogeneity of age and gender was assessed using the Mann–Whitney U test.

Pearson’s Chi-squared test was performed to determine the difference in the prevalence of AIS, STEMI, and other investigated cardiovascular diseases between the RAO group and the control group. The two-proportion Z-test was used to examine the role of age and gender in RAO patients who developed AIS/STEMI within 10 years. To examine differences in AIS and STEMI risk, univariate logistic regression was used; in all cases, we adjusted for age and gender. Similarly, the role of HT, T1DM, T2DM, AF, hyperlipidemia, and IHD in AIS and STEMI development was examined in comparison with the controls. Multiple logistic regression analysis was used to investigate how multiple risk factors influence the development of AIS and STEMI in RAO patients.

Statistical analyses were performed using Intercooled Stata v17 (StataCorp. 2021. Stata Statistical Software: Release 17. College Station, TX, USA: StataCorp LLC). All results with *p* < 0.05 were considered statistically significant.

## 3. Results

The demographic data of the RAO and control groups are summarised in Table 1.

Data from 169 patients with RAO (106 with CRAO and 63 with BRAO) were compared with data from 169 controls without any retinal vascular disease. Our data showed a non-normal distribution with the Shapiro–Wilk test (*p* < 0.05). There were significantly more men in the RAO group than women (*p* = 0.007). The age and gender of RAO and control patients were homogeneous as per the Mann–Whitney U test. The average age in the RAO group was 64.18 ± 10.00 years, compared to 63.88 ± 10.43 years in the control group.

During the study period, the retinal vascular disease became bilateral in two patients: one developed CRAO, and the other developed BRAO in both eyes. Of the 169 RAO patients, 40 developed AIS and 32 had STEMI, representing a significant increase compared to the control group according to Pearson’s Chi-squared test (*p*1 < 0.001 and *p*2 < 0.001, respectively).

Regarding comorbidities, which are also cardio/cerebrovascular risk factors, HT and hyperlipidaemia was found to be significantly more frequent in the RAO group than in the control group according to Pearson’s Chi-squared test (*p*1 = 0.005, *p*2 < 0.001). No significant difference was found between the two groups for T1DM, T2DM, AF, and IHD (*p*1 = 0.346, *p*2 = 0.096, *p*3 = 0.309, and *p*4 = 0.641, respectively). Among 169 RAO patients, 32 cases of AF were found.

### 3.1. Differences in the Prevalence of AIS and STEMI between Age Groups

We used the two-proportion Z-test to examine how the number of AIS and STEMI cases increases with age in 10-year intervals. We also analysed the role of advancing age as a risk factor and its pathogenetic role. We concluded that there is a large increase in the prevalence of both AIS and STEMI over the age of 50. In those aged between 60 and 70 years, the incidence of AIS and STEMI remains high, without major further changes. The difference in AIS and STEMI was significant between the two age groups (under and over 50 years of age; *p*1 < 0.001 and *p*2 < 0.001, respectively) but not significant in proportion between the RAO group and the control group.

### 3.2. Risk-Enhancing Effect of RAO towards AIS and STEMI

Univariate logistic regression analysis was used to analyse the risk of RAO and comorbidities for AIS and STEMI. For AIS, RAO represents an 8.18-fold increase in risk (OR: 8.18, *p* < 0.001, 95% CI: 3.09–21.64; Table 2).

For STEMI, RAO represents a 3.10-fold increase in risk (OR: 3.10, *p* = 0.007, 95% CI: 1.36–7.08). In addition, T1DM represents a 3.51-fold increase in risk (OR: 3.51, *p* = 0.013, 95% CI: 1.31–9.42); AF, a 2.35-fold increase in risk (OR: 2.35, *p* = 0.030, 95% CI: 1.08–5.07); hyperlipidaemia, a 2.30-fold increase in risk (OR: 2.30, *p* = 0.023, 95% CI: 1.12–4.72); and IHD represents a 2.13-fold increase (OR: 2.13, *p* = 0.044, 95% CI: 1.02–4.46). The differences between RAO patients and controls were also significant for these risk factors (Table 3).

### 3.3. Impact of the Co-Appearance of Cardiovascular Diseases

We used multiple logistic regression analysis to investigate how the co-appearance of different risk factors/cardiovascular diseases influences the development of AIS and STEMI in RAO patients. The risk of developing AIS increases 8.18-fold for RAO patients (OR: 8.18, *p* < 0.001, 95% CI: 3.09–21.64). When combined with RAO, the presence of HT elevates the odds ratio to 11.65 (OR: 11.65, *p* < 0.001, 95% CI: 4.44–30.55). AF contributes to a 4.36-fold increased risk (OR: 4.36, *p* = 0.001, 95% CI: 1.83–10.34), while hyperlipidaemia shows a 2.66-fold increased risk (OR: 2.66, *p* = 0.009, 95% CI: 1.28–5.53). Additionally, if T1DM (OR: 25.11, *p* < 0.001, 95% CI: 4.86–129.71), T2DM (OR: 15.38, *p* < 0.001, 95% CI: 4.80–49.24), or hyperlipidaemia (OR: 2.99, *p* = 0.004, 95% CI: 1.43–6.25) is present as a third factor, the risk of AIS remains significantly elevated. The presence of IHD alongside RAO and HT does not increase the risk further (OR: 10.04, *p* < 0.001, 95% CI: 3.26–30.94; Table 4).

The risk of developing STEMI increases 3.10-fold for RAO patients (OR: 3.10, *p* = 0.007, 95% CI: 1.36–7.08). If HT is present along with RAO, the odds ratio increases to 4.93 (OR: 4.93, *p* < 0.001, 95% CI: 2.24–10.85). The presence of AF alongside RAO results in a 4.92-fold higher risk (OR: 4.92, *p* < 0.001, 95% CI: 2.09–11.56). Hyperlipidaemia combined with RAO is associated with a 3.34-fold increased risk (OR: 3.34, *p* = 0.002, 95% CI: 1.56–7.15). Furthermore, if T1DM (OR: 4.22, *p* = 0.013, 95% CI: 1.36–13.15), T2DM (OR: 4.79, *p* = 0.005, 95% CI: 1.60–14.37), or hyperlipidaemia (OR: 3.62, *p* = 0.001, 95% CI: 1.69–7.78) is present as a third factor, there is no additional increase in the risk. The co-existence of IHD with RAO and HT increases the odds ratio of developing STEMI even more than the preceding factors (OR: 5.52, *p* = 0.001, 95% CI: 2.10–14.53; Table 5).

### 3.4. Mortality

Unfortunately, 23 RAO patients died during the study period due to cardio/cerebrovascular complications. We found no difference in the frequencies of AIS- and STEMI-related deaths (Table 1). We found no significant difference in the temporal relationship between AIS and RAO in the RAO group; 13 patients had cerebrovascular events before RAO and 12 had them after RAO. However, no data were available on the time of ischaemic stroke onset for 15 patients. In addition, we found no significant difference in the temporal relationship between STEMI and RAO in the RAO group; STEMI developed in 19 patients before RAO and in 6 after RAO. However, no data were available on the time of onset of STEMI for seven patients.

## 4. Discussion

The blood supply to the retina is ensured by the central retinal artery, which is the first branch of the ophthalmic artery. The ophthalmic artery originates from a branch of the internal carotid artery. The retinal tissue is sustained by a dual vascular supply. The central retinal artery enters the eye through the optic nerve and branches out to supply blood to the inner two-thirds of the retina. It then branches inside the retina, providing perfusion to the innermost layers, including the nerve fibre layer, the ganglion cell layer, and the inner plexiform layer. The choroidal vasculature provides the vascular supply to the outer layers of the retina, especially the photoreceptor layer. The choroid is supplied by the posterior ciliary arteries, which arise from the ophthalmic artery. The convergence of these circulatory axes serves to maintain the metabolic needs of the retinal milieu, thereby maintaining the vital physiological function of the retina. Retinal artery occlusion (RAO) can occur due to the occlusion of the central retinal artery caused by circulatory disturbance or embolus. Emboli most commonly originate from calcified plaques of the atherosclerotic carotid artery, but cholesterol embolism and thromboembolism are also common. The development of arterial plaques is influenced by factors such as hypertension, smoking, elevated cholesterol levels, and potential inflammatory processes within the inner layer of the arteries, leading to endothelial damage. This results in increased vascular permeability, initiating the accumulation of lipid and inflammatory components within the arterial wall through activated immune cascade mechanisms. Macrophages also participate in this process, leading to the formation of lipid-filled foam cells. In advanced stages, arterial plaques develop within the vessel walls, from which tiny emboli can obstruct the ophthalmic artery. Oxidative stress leads to an imbalance between antioxidants and free radicals, which damage normal cells and can also affect the vascular walls. Damage to the vessel wall can lead to atherosclerosis. As a result of vasoconstriction, the process worsens and the blood supply to the eye is disrupted [18].

Arteriosclerosis plays a fundamental role in the development of RAO, AIS, and STEMI. With progressive age, multiple cardiovascular risk factors can lead to plaque formation in the aorta and the internal carotid artery or its branches, parts of which can break off, leading to the development of acute RAO, AIS, or STEMI through embolisation [21,22,23,24].

In our retrospective cohort study (10-year interval), we investigated the role of age, gender, HT, T1DM/T2DM, AF, hyperlipidaemia, and IHD in RAO development. When analysing the role of age, we found that RAO, AIS, and STEMI rarely developed in people under 40 years of age. However, over 50 years of age, the incidence of AIS and STEMI increased in the RAO group. Further statistical analysis comparing RAO and non-RAO cases in terms of age revealed a strongly significant difference in those under and over 50 years of age [25]. The pathogenetic role of age in ischaemic stroke was demonstrated by Salinero-Fort and colleagues. Their study in a Mediterranean population of RAO patients revealed age to be the only independent factor in the development of cardio/cerebrovascular events [26,27].

In contrast to our results, a Korean study group only found a significant increase in stroke in RAO patients over 65 years of age [28]. The difference may be because the Hungarian population has a higher incidence of cardiovascular disease compared to the Korean population. While the genetic makeup of the Hungarian and Asian populations is different, the disparity may also be due to differences in diet [29].

In our study, men developed RAO at a significantly higher rate than women. However, Jeong et al. found no significant difference between genders [30]. According to our study, among men at their prime strength and working capacity, there was a higher likelihood of developing RAO or acute cardio/cerebrovascular events, which significantly impacted their quality of life.

The incidence of AF in patients with RAO is high. RAO patients with AF are significantly more likely to develop STEMI, although no significant increase in risk was found for AIS. AF most commonly develops as a result of the dilatation of the right atrium. A thrombus may develop in the right atrial auricle, which in the case of an open foramen ovale may even lead to embolisation, often resulting in ischaemic stroke or arterial occlusion [31]. Arrhythmias may be followed by microthrombus formation, which may also play a role in the development of AIS and RAO. In accordance with several authors, we found that RAO patients had significantly higher rates of hypertension and T2DM [32,33]. Similar to our results, Rim et al. found an association between hypertensive retinopathy and cerebrovascular disease [26]. Other authors have reported that only atherosclerosis of the large arteries plays a role in the development of RAO and stroke. The same authors found no significant differences between genders or cardiovascular risk factors in this regard [30].

Hyperlipidaemia can play a role in the development of progressive atherosclerosis. In untreated hyperlipidaemia, plaques can form in the walls of blood vessels, affecting large vessels such as the intracranial vessels, the coronary artery, and also smaller vessels such as the central retinal artery. Excess circulating low-density lipoprotein cholesterol promotes the accumulation of cholesterol within arterial walls, triggering an inflammatory response and the formation of atherosclerotic plaques. These plaques can eventually lead to arterial stenosis and contribute to the development of cardiovascular diseases, such as coronary artery disease and stroke. In our patients with hyperlipidaemia associated with HT, the odds ratio for the development of AIS and STEMI was significantly increased [34].

In agreement with other authors, we found that RAO patients developed AIS/STEMI significantly more frequently within 10 years of follow-up. Park et al. found a higher rate of ischaemic stroke within 1 week after RAO [35]. Chang et al. found a higher rate of stroke within 3 years after the onset of CRAO [36]. The American Heart Association considers CRAO a form of ischaemic stroke and recommends concordant acute treatment [37]. Laczynski et al., in contrast to our results, found no association between the development of RAO and stroke. The authors’ follow-up time, compared to our 10-year follow-up time, was only 2.2 years on average [38].

The main benefit of our work is the analysis of the co-occurrence of multiple risk factors within the RAO group. We demonstrated that the accumulation of one, two, or possibly three risk factors/cardiovascular diseases together increased the odds of developing AIS by a factor of up to 25 and those of developing STEMI by a factor of up to 5. Based on our results and given that the first examiner of RAO patients is often the ophthalmologist, some RAO, AIS, and STEMI cases can be prevented among these cardiovascularly burdened patients with a proper general medical examination and the treatment of accompanying diseases. Cooperation between ophthalmologists, neurologists, and cardiologists can prevent RAO and possibly also AIS/STEMI.

As the first point of care for RAO patients, we should consider the increased risk of AIS/STEMI development. The patient should be referred to neurology/emergency care as soon as possible, where appropriate investigation (carotid Doppler, blood pressure adjustment, magnetic resonance imaging) and secondary prevention (e.g., clopidogrel, aspirin) can prevent the development of severe pathologies in some cases [39,40,41,42,43,44].

The main limitation of our study is that although we analysed patients’ medical records using a national electronic database in addition to using our clinic’s examination data, we were unable to access all patients’ records due to patient digital self-determination. A further difficulty was that the documentation was sometimes incomplete and did not always cover the risk factors under investigation. In the future, we plan to conduct more extensive and prolonged studies with a larger patient cohort that could provide a more comprehensive understanding of the interplay between RAO, AF, and the risk of AIS and STEMI. We also plan to collaborate with neurologist colleagues to analyse carotid Doppler scans and investigate the role of ACI stenosis in RAO patients.

Considering the large number of patients and the long follow-up time, it is hoped that the abovementioned difficulties will not significantly affect the results. Furthermore, drawing the control group from the same institutional environment serves to control for certain confounding variables that might arise from differing institutional practices, patient demographics, or geographic factors.

## 5. Conclusions

RAO is associated with an increased risk of AIS and STEMI, especially in patients with multiple cardiovascular risk factors. The early detection, proper medical examination, and treatment of accompanying diseases, through collaboration between ophthalmologists, neurologists, and cardiologists, can potentially prevent RAO, AIS, and STEMI in cardiovascularly burdened patients. It is crucial to recognise increased risk and promptly refer patients for appropriate investigations and secondary prevention measures. The use of clopidogrel or aspirin treatment can be beneficial for the secondary prevention of the development of RAO in the fellow eye, AIS/STEMI, provided that no contraindication arises.

## Figures and Tables

**Figure 1 medicina-59-01680-f001:**
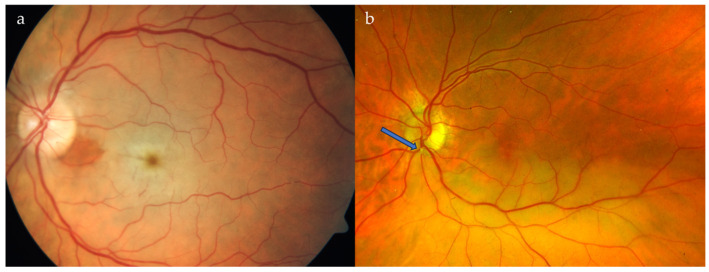
(**a**) Colour fundus photograph showing characteristic findings of central retinal artery occlusion (pale retina due to oedema, cherry-red spot in the foveolar area). (**b**) Branch retinal artery occlusion—the pale retina is visible in the region supplied by the temporal inferior arterial branch. The embolus causing the blockage is indicated by the blue arrow.

**Table 1 medicina-59-01680-t001:** Demographic data, cardiovascular risk factors/diseases, and mortality of the retinal artery occlusion (RAO) and control patients (including the central retinal artery occlusion (CRAO) and branch retinal artery occlusion (BRAO) subgroups).

	RAO	CRAO	BRAO	Control
Number of patients	169	106	63	169
Both eyes	2 (1.18%)	1 (0.94%)	1 (1.59%)	n.a. *^6^
Mean age	64.18 ± 10.00	64.69 ± 10.48	63.33 ± 9.17	63.88 ± 10.43
Gender (male/female)	97/72 (57.40%/42.60%)	67/39 (63.21%/36.79%)	30/33 (47.62%/52.38%)	97/72 (57.40%/42.60%)
AIS *^2^	40 (23.67%)	25 (23.58%)	15 (23.81%)	5 (2.96%)
AIS before RAO	13 (7.69%)	9 (8.49%)	4 (6.35%)	n.a.
Average duration (years ± SD *^7^)	4.60 ± 4.17	5.24 ± 4.60	3.17 ± 3.00	n.a.
RAO before AIS	12 (7.10%)	9 (8.49%)	3 (4.76%)	n.a.
Average duration (years ± SD)	1.58 ± 1.90	1.21 ± 1.62	2.66 ± 2.66	n.a.
STEMI *^8^	32 (18.93%)	20 (18.87%)	12 (19.05%)	9 (5.33%)
STEMI before RAO	19 (11.24%)	10 (9.43%)	9 (14.29%)	n.a.
Average duration (years ± SD)	13.63 ± 7.90	10.68 ± 7.92	16.90 ± 6.86	n.a.
RAO before STEMI	6 (3.55%)	5 (4.72%)	1 (1.59%)	n.a.
Average duration (years ± SD)	3.11 ± 10.05	2.65 ± 9.15	5.41	n.a.
HT *^4^	145 (85.80%)	88 (83.02%)	57 (90.48%)	124 (73.37%)
T1DM *^9^	18 (10.65%)	14 (13.21%)	4 (6.35%)	13 (7.69%)
T2DM *^10^	38 (22.49%)	26 (24.53%)	12 (19.05%)	26 (15.38%)
AF *^1^	32 (18.93%)	20 (18.87%)	12 (19.05%)	25 (14.79%)
HL *^3^	56 (33.14%)	30 (28.30%)	26 (41.27%)	21 (12.43%)
IHD *^5^	52 (30.77%)	27 (25.47%)	25 (39.68%)	56 (33.14%)
Mortality	23 (13.61%)	14 (13.21%)	9 (14.29%)	n.a.

*^1^ AF = atrial fibrillation, *^2^ AIS = acute ischaemic stroke, *^3^ HL = hyperlipidemia, *^4^ HT = hypertension, *^5^ IHD = ischaemic heart disease, *^6^ n.a. = not applicable, *^7^ SD = standard deviation, *^8^ STEMI = ST-elevation myocardial infarction, *^9^ T1DM = type 1 diabetes mellitus, *^10^ T2DM = type 2 diabetes mellitus.

**Table 2 medicina-59-01680-t002:** Univariate logistic regression to analyse how the risk of developing acute ischaemic stroke (AIS) is increased in the presence of certain cardiovascular risk factors/diseases. Significant values are in bold.

	Odds Ratio	*p*	95% CI *
RAO	**8.18**	**<0.001**	3.09–21.64
HT	4.09	0.065	0.92–18.27
T1DM	1.12	0.831	0.39–3.25
T2DM	1.25	0.578	0.57–2.77
AF	1.66	0.203	0.76–3.60
HL	1.85	0.085	0.92–3.72
IHD	0.73	0.412	0.35–1.55

* 95% CI = 95% confidence interval.

**Table 3 medicina-59-01680-t003:** Univariate logistic regression to analyse how the risk of developing ST-elevation myocardial infarction (STEMI) is increased in the presence of certain cardiovascular risk factors/diseases. Significant values are in bold.

	Odds Ratio	*p*	95% CI
RAO	**3.10**	**0.007**	1.36–7.08
HT	7.01	0.064	0.90–54.87
T1DM	**3.51**	**0.013**	1.31–9.42
T2DM	1.12	0.798	0.48–2.58
AF	**2.35**	**0.030**	1.08–5.07
HL	**2.30**	**0.023**	1.12–4.72
IHD	**2.13**	**0.044**	1.02–4.46

**Table 4 medicina-59-01680-t004:** Multiple logistic regression to analyse how the co-existence of cardiovascular diseases in RAO patients affects the development of AIS. Significant values are in bold.

	Odds Ratio	*p*	95% CI
RAO	**8.18**	**<0.001**	3.09–21.64
RAO + AF	**4.36**	**0.001**	1.83–10.34
RAO + HL	**2.66**	**0.009**	1.28–5.53
RAO + HT	**11.65**	**<0.001**	4.44–30.55
RAO + HT + T1DM	**25.11**	**<0.001**	4.86–129.71
RAO + HT + T2DM	**15.38**	**<0.001**	4.80–49.24
RAO + HT + IHD	**10.04**	**<0.001**	3.26–30.94
RAO + HT + HL	**2.99**	**0.004**	1.43–6.25

**Table 5 medicina-59-01680-t005:** Multiple logistic regression to analyse how the co-existence of cardiovascular diseases in RAO patients affects the development of STEMI. Significant values are in bold.

	Odds Ratio	*p*	95% CI
RAO	**3.10**	**0.007**	1.36–7.08
RAO + AF	**4.92**	**<0.001**	2.09–11.56
RAO + HL	**3.34**	**0.002**	1.56–7.15
RAO + HT	**4.93**	**<0.001**	2.24–10.85
RAO + HT + T1DM	**4.22**	**0.013**	1.36–13.15
RAO + HT + T2DM	**4.79**	**0.005**	1.60–14.37
RAO + HT + IHD	**5.52**	**0.001**	2.10–14.53
RAO + HT + HL	**3.62**	**0.001**	1.69–7.78

## Data Availability

All data are included in this manuscript.

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
