# Peer review of "Analysis of the Association between Retinal Artery Occlusion and Acute Ischaemic Stroke/ST-Elevation Myocardial Infarction and Risk Factors in Hungarian Patients"

_medicina, 2023, doi:10.3390/medicina59091680_

Round 1

Reviewer 1 Report

Minor improvement.

Author Response

Szabolcs Balla MD

Nagyerdei blvd 98

4032 Debrecen, Hungary

szabb12@gmail.com

+36-20/588-0680

2023.08.29

Reviewer 1

Subject: Response to Reviewer 1's Comments - Manuscript ID [medicina-2569976]

Dear Reviewer 1,

We sincerely appreciate your thoughtful evaluation of our manuscript titled "Analyzing Correlations of Retinal Artery Occlusion with Acute Ischemic Stroke and ST-Elevation Myocardial Infarction" (Manuscript ID: [medicina-2569976]) submitted to MDPI Journal "Medicina." We are grateful for the time and effort you dedicated to the review process, and we would like to assure you that we have carefully considered each of your comments and recommendations.

Below, we address each of your specific comments:

  1. In introduction, line 36, replace circulatory with vascular.

The term has been corrected.

  1. Too many small paragraphs in the introduction section. Try to have maximum three paragraphs in the introduction.

The paragraphs have been reduced to the proposed number.

  1. Identifiers like the place of the study should not be divulged in the methods section.

The study location has been removed from the methods section, as you requested.

  1. The 169 healthy subjects were included with the criteria that they had no history of any previous retinal disorder. However, was an ophthalmic examination was done in such patient to ensure that they had no retinal pathology?

Patients who were about to undergo cataract surgery and who did not have any retinal vascular disease in the electronic database were selected as controls. All preoperative patients underwent a complete ophthalmic examination in our clinic before surgery. For clarity, we also outlined the above in the manuscript.

  1. Lines 116-117, there were significantly more men in the RAO group (p< 0.05). However, in the table 1, both RAO and control group have 97 males. So how can it be significant if both had the same numbers? Please provide an explanation.

We deliberately chose age- and gender-matched control groups, indeed there were equal numbers of men and women in the patient and control groups. The statement suggests that there are significantly more men than women in the RAO group. Given the ambiguity of the sentence, we have completed it as follows: “There were significantly more men in the RAO group than women (p = 0.007).”

  1. The average duration of development of either AIS/STEMI following a RAO should be mentioned in the results.

As proposed, these data have been added to Table 1.

  1. Lines 132-133, We concluded that over 50 years of age, there is a sudden increase in both AIS and STEMI prevalence. What do the authors mean by sudden? Sudden is qualitative. Kindly provide a quantitative value.

The word "sudden" is indeed incorrect, we have corrected the text to "large increase in the number of cases". AIS developed in 1 patient under 50 years and in 39 patients over 50 years. For STEMI, 1 patient under 50 and 31 patients over 50 developed the disease.

  1. What is the average duration of follow up for cases and controls? Have the control patients also been followed up.

This is a cohort study of RAO patients who developed RAO between 2009-2019, so the follow-up time after RAO does vary from patient to patient. An age and gender matched control group was chosen for comparison.

  1. Lines 191-196 in discussion are basically a repeat of what has already been mentioned in the introduction. Can be avoided.

Repetitive part is deleted from the manuscript. We start the “Discussion” section with the pathophysiology of RAO.

  1. “According to the results of our work group, men at the peak of their strength and work capacity were more likely to develop RAO or an acute cardio/cerebrovascular event that had a major impact on their quality of life”. The mean age of the patients in the RAO group is 64.18 ± 10.00. It is very hard to conceive how patients in this age group can be considered to be at the peak of their strength and work capacity. In many countries, 60 years is the retirement age.

In our country, the retirement age is 65, which may currently be subject to increase. While "peak of their strength" might not be the best phrasing, we believe that this age group is still in their prime and often at the height of their careers. The term has been changed in the manuscript as follows: “According to our study, among men in their prime strength and working capacity, there was a higher likelihood of developing retinal artery occlusion (RAO) or acute cardio/cerebrovascular events, which significantly impacted their quality of life.”

  1. It is quite surprising that deranged lipid profile has not been included in the study as a risk factor. Similarly, deranged serum homocysteine is also an important risk factor which could have been analysed.

Data on hypercholesterolaemia in RAO patients were recorded in our outpatient clinic. Among the 169 patients, 45 had hypercholesterolemia. However, in the electronic database from which we selected the controls, cases of hypercholesterolemia were incompletely available. Therefore, we were unable to analyze this in comparison to the control group and had to omit it from our study.

Often, patients even deny smoking despite being heavy smokers. Determining smoking status is very challenging due to the limited reliable data available in the Hungarian electronic database.

Analysis of homocysteine, which plays a role in thrombophilia and atherosclerosis development, was not performed in our current study. We plan to conduct more comprehensive investigations in the future.

  1. The manuscript appears monotonous to read. Kindly provide some pictures in the form of coloured graphs.

Odds ratios are also presented in the form of 4 coloured graphs.

  1. The manuscript is filled with multiple small paragraphs. This does not look good at all. Combine and decrease the number of paragraphs.

We have reduced the number of paragraphs as proposed. Thank you for your constructive comment!

  1. Reference 1 seems to be incomplete.

The reference has been updated.

Thank you for your dedication to the peer review process and your valuable input. We believe that your feedback has played a crucial role in strengthening our manuscript. We are grateful for your time and expertise!

Sincerely,

Szabolcs Balla MD

University of Debrecen, Department of Ophthalmology, Debrecen, Hungary

Reviewer 2 Report

This is an interesting study, but there are some problems in this manuscript.

The ophthalmic artery (OA) was a branch of the internal carotid artery. the occlusion of OA will cause visual loss. It is a lack of data about the carotid artery and intima-media thickness (IMT)of the common carotid artery.

IMT was a marker of atherosclerosis. The RAO may be due to related atherosclerosis that cause the OR was not related to atherosclerosis-relative factors.

The most common cause of RAO is embolization in Discussion, atrial fibrillation (Af) was the most common cause of ischemic stroke. The study is lack Af data.

The logistic regression needs to adjust the age and gender.

Author Response

Subject: Response to Reviewer 2's Comments - Manuscript ID [medicina-2569976]

Dear Reviewer 2,

We sincerely appreciate your thoughtful evaluation of our manuscript titled "Analyzing Correlations of Retinal Artery Occlusion with Acute Ischemic Stroke and ST-Elevation Myocardial Infarction" (Manuscript ID: [medicina-2569976]) submitted to MDPI Journal "Medicina." We are grateful for the time and effort you dedicated to the review process, and we would like to assure you that we have carefully considered each of your comments and recommendations.

Below, we address each of your specific comments:

The ophthalmic artery (OA) was a branch of the internal carotid artery. the occlusion of OA will cause visual loss. It is a lack of data about the carotid artery and intima-media thickness (IMT) of the common carotid artery.

You're absolutely correct. Following the onset of retinal artery occlusion (RAO), our established protocol guides us in referring patients to the Neurology Department. There, a comprehensive series of assessments - such as Carotid Doppler ultrasound and brain MRI - are conducted to assess the patient’s cerebrovascular status.

IMT was a marker of atherosclerosis. The RAO may be due to related atherosclerosis that cause the OR was not related to atherosclerosis-relative factors.

We appreciate the reviewer's keen observations. Our study was primarily focused on the connection of RAO and AIS/STEMI. We acknowledge that the carotid artery status and IMT could provide valuable context, but as of the current study's scope, we did not include data on carotid artery stenosis. However, we fully recognise the importance of this aspect and, as modified in the manuscript, we plan to conduct more comprehensive studies in the future that specifically investigate carotid stenosis and its association with the outcomes observed in the present study.

The most common cause of RAO is embolization in Discussion, atrial fibrillation (Af) was the most common cause of ischemic stroke. The study is lack Af data.

According to the literature AF is undoubtedly the cause of most strokes. Out of a total of 169 RAO patients, atrial fibrillation occurred in 32 patients. This data has been integrated into the "Results" section. Given the importance of this data in contributing to a comprehensive understanding, we sincerely appreciate your suggestion. In subsequent endeavors, we will place special emphasis on providing information about the primary risk factors.

The logistic regression needs to adjust the age and gender.

Logistic regression analysis was adjusted for age and gender in all cases. For the sake of completeness, we have added this information to the “Statistical analysis” section.

In conclusion, we are grateful to the reviewer for the valuable feedback and thoughtful critique of our manuscript. The reviewer's insights will undoubtedly guide us in improving the quality and clarity of our future research. Should the reviewer have any further suggestions or inquiries, we are more than willing to engage in further discussion.

Sincerely,

Szabolcs Balla MD

University of Debrecen, Department of Ophthalmology, Debrecen, Hungary

Reviewer 3 Report

I would like to congratulate the authors on their interesting work. A few issues that need be addressed are the following:

1.In the methods section in line 59-62 181 RAO patinets were identified of which 169 were included in the analysis. Why were the 12 patients excluded? Could you provide a flow chart of the selection proccess e.g. 181 patients identified, how many met exclusion criteria how many did not have the parameters of interest documented etc.

2. Why didn't you include hypercholesterolemia or smoking to your analysis?

3.In the discussion section could you discuss in detail the pathophysiology of retinal artery occlusion, endothelial dysfunction, oxidative stress, plaque formation that you briefly mention in lines 196-201 so that the ophtalmology audience could better understant those mechanisms?

No comments.

Author Response

Subject: Response to Reviewer 3's Comments - Manuscript ID [medicina-2569976]

Dear Reviewer 3,

We sincerely appreciate your thoughtful evaluation of our manuscript titled "Analyzing Correlations of Retinal Artery Occlusion with Acute Ischemic Stroke and ST-Elevation Myocardial Infarction" (Manuscript ID: [medicina-2569976]) submitted to MDPI Journal "Medicina." We are grateful for the time and effort you dedicated to the review process, and we would like to assure you that we have carefully considered each of your comments and recommendations.

Below, we address each of your specific comments:

1.In the methods section in line 59-62 181 RAO patinets were identified of which 169 were included in the analysis. Why were the 12 patients excluded? Could you provide a flow chart of the selection proccess e.g. 181 patients identified, how many met exclusion criteria how many did not have the parameters of interest documented etc.

1 patient had concomitant wet age-related macular degeneration, 1 patient had RAO after coronary bypass, 1 patient had RAO combined with anterior ischaemic optic neuropathy, 2 patients had RAO combined with BRVO and 7 patients had insufficient documentation. In line with the reviewer's proposal the inclusion and exclusion criteria are explained in more detail in the manuscript.

  1. Why didn't you include hypercholesterolemia or smoking to your analysis?

Regarding hypercholesterolemia, we did collect data in our specialty clinic within the RAO patient group. Out of the 169 patients included, we found records for 45 patients with hypercholesterolemia. However, it's important to note that within the electronic database from which we selected control participants, information about hypercholesterolemia were only minimally present. This limited presence in the control group prevented us from conducting a meaningful comparative analysis, leading us to omit this aspect from our study.

As for smoking, it's noteworthy that many patients, even heavy smokers, tend to deny their smoking habit. Determining smoking status proves to be challenging, particularly due to the scarcity of reliable data available in the Hungarian electronic database concerning this specific aspect.

To overcome these limitations of the present study, we see the solution in prospectively collecting data in both the patient and control groups in our future similar comparisons.

3.In the discussion section could you discuss in detail the pathophysiology of retinal artery occlusion, endothelial dysfunction, oxidative stress, plaque formation that you briefly mention in lines 196-201 so that the ophtalmology audience could better understant those mechanisms?

As suggested, the discussion was extended.

We would like to express our gratitude for your dedication to the peer review process and your constructive input. Your diligence in addressing these critical aspects has undoubtedly contributed to the improvement of the manuscript. Thank you for your time, expertise, and commitment to advancing scientific discourse. We are truly grateful!

Sincerely,

Szabolcs Balla MD

University of Debrecen, Department of Ophthalmology, Debrecen, Hungary

Round 2

Reviewer 1 Report

Dear authors,

Thank you for modifying your paper. By adding color images, I did not mean that you had to colour the graphs. What I meant is to provide some clinical images of patients with retinal artery occlusion. It will be good if you can provide fundus photos.

Author Response

Szabolcs Balla MD

Nagyerdei blvd 98

4032 Debrecen, Hungary

szabb12@gmail.com

+36-20/588-0680

2023.09.04

Reviewer 1

Subject: Response to Reviewer 1's Round 2 Comments - Manuscript ID [medicina-2569976]

Dear Reviewer 1,

Thank you again for your thoughtful review of our manuscript "Analyzing Correlations of Retinal Artery Occlusion with Acute Ischemic Stroke and ST-Elevation Myocardial Infarction" (Manuscript ID: [medicina-2569976]), submitted to the MDPI Journal "Medicina". Sorry for the misunderstanding, we have corrected the manuscript as follows:

  • Thank you for modifying your paper. By adding color images, I did not mean that you had to colour the graphs. What I meant is to provide some clinical images of patients with retinal artery occlusion. It will be good if you can provide fundus photos.
  • As recommended, we have removed the color graphs and incorporated fundus photographs of patients with central and branch retinal occlusions into the manuscript.

Thank you again for your dedication to the peer review process and your valuable input. We are grateful for your time and expertise!

Sincerely,

Szabolcs Balla MD

University of Debrecen, Department of Ophthalmology, Debrecen, Hungary

Reviewer 2 Report

The authors answer some questions, but it still has some problems as follows:

It needs to perform the adjusted risk of ischemic stroke and MI in AF.

The carotid stenosis was related to ischemic stroke, the finding in Carotid Doppler ultrasound and brain MRI needed to be shown. 

Atherosclerosis related to hypertension, diabetes mellitus, and hyperlipidemia, the authors need to provide the combined risk factors to compare with RAO risk.

Author Response

Szabolcs Balla MD

Nagyerdei blvd 98

4032 Debrecen, Hungary

szabb12@gmail.com

+36-20/588-0680

2023.09.04

Reviewer 2

Subject: Response to Reviewer 2's Round 2 Comments - Manuscript ID [medicina-2569976]

Dear Reviewer 2,

We sincerely appreciate your thoughtful evaluation of our manuscript titled "Analyzing Correlations of Retinal Artery Occlusion with Acute Ischemic Stroke and ST-Elevation Myocardial Infarction" (Manuscript ID: [medicina-2569976]) submitted to MDPI Journal "Medicina." We value your efforts and the time you dedicated to our manuscript.

Below, we address each of your specific comments:

  1. It needs to perform the adjusted risk of ischemic stroke and MI in AF.

We employed Pearson's Chi-squared test to compare hyperlipidemia and AF between the RAO and control groups. While there was a significant difference in the former, the latter did not yield significance. Furthermore, univariate logistic regression analysis revealed that AF and hyperlipidemia did not significantly increase the odds ratio for stroke; however, they were both significant factors for STEMI. We have incorporated these findings into the manuscript as recommended.

  1. The carotid stenosis was related to ischemic stroke, the finding in Carotid Doppler ultrasound and brain MRI needed to be shown. 

We agree that carotid Doppler and brain MRI are important in AIS and RAO patients. A detailed analysis of these is not included in the objectives of our manuscript. We aspire to collaborate with neurologists in future endeavors to explore and analyse such data within the broader context of our research.

  1. Atherosclerosis related to hypertension, diabetes mellitus, and hyperlipidemia, the authors need to provide the combined risk factors to compare with RAO risk.

At your suggestion, we performed the combined risk factor analysis on our RAO patients. We have also incorporated the outcomes into the manuscript.

We would like to express again our sincere gratitude for your valuable comments and suggestions regarding our manuscript. We have incorporated the suggested modifications and additions into the manuscript and hope that it now meets the acceptability criteria.

Sincerely,

Szabolcs Balla MD

University of Debrecen, Department of Ophthalmology, Debrecen, Hungary
